# Defense against Adversarial Examples by Encoder-Assisted Search in the Latent Coding Space

## Abstract

Deep neural networks were shown to be vulnerable to crafted adversarial perturbations, and thus bring serious safety problems. To solve this problem, we proposed AE-GAN$_{+rs}$, a framework for purifying input images by searching the closest natural reconstruction with little computation. We first build a reconstruction network AE-GAN, which adapted auto-encoder by introducing adversarial loss to the objective function. In this way, we can enhance the generative ability of decoder and preserve the abstraction ability of encoder to form a self-organized latent space. In the inference time, when given an input, we will start a search process in the latent space which aims to find the closest reconstruction to the given image on the distribution of natural data. The encoder can provide a good start point for the searching process, which saves much computation cost. Experiments show that our method is robust against various attacks and can reach comparable even better performance to similar methods with much fewer computations.

## 1 Introduction

Deep neural networks (DNNs) have achieved state-of-the-art performances among various challenging computer vision tasks. However, they are shown to be vulnerable to adversarial attacks (Szegedy et al., 2013; Goodfellow et al., 2014b; Madry et al., 2017b), which add carefully crafted perturbations to a legitimate input sample. The perturbations are small and not perceptible to a human, but they can significantly mislead the target model. Moreover, adversarial examples have considerable transferability between different models, which entertains the feasibility of black-box attacks in the real world. Therefore, it is important to find effective strategies to defend against adversarial attacks.

There have been many efforts to defend against adversarial attacks by diminishing perturbations of samples before feeding them into classifiers. Auto-encoder based methods have achieved prevalence though (Gu & Rigazio, 2014; Meng & Chen, 2017), they are shown to be vulnerable to white-box attacks (Samangouei et al., 2018) because the model stacked by auto-encoder and classifier can be attacked again. Other generative models (Samangouei et al., 2018; Song et al., 2017) have been trained to model the distribution of normal data and project back the adversarial samples to the manifold of normal samples. In the process of purifying input images, they often include complicated optimization. Therefore, they can easily resist the white-box attacks but at the cost of time.

Compared to natural images, adversarial examples have much lower probability densities under the image distribution (Song et al., 2017), which is regarded as an unknown input pattern for an auto-encoder trained with legitimate examples. As a result, the output of the encoder can be influenced by the crafted perturbations and lead to a top-down reconstruction far away from the input. When humans handle unknown input patterns, we will first find a conception in our mind and correct our conception step by step by checking the consistency between the input and the reconstruction. We can apply this mechanism to neural networks, where the latent code for reconstruction is calculated by a search process guided by feedback from the visual space. The same idea has been included in Defense-GAN (Samangouei et al., 2018), where a generator is trained to model the distribution of unperturbed images. When given an image, it uses GD minimization to find the optimum latent code, which corresponds to the closest output without perturbations to the given image. However, Defense-GAN requires multiple random restart points and considerable iterations to guarantee the

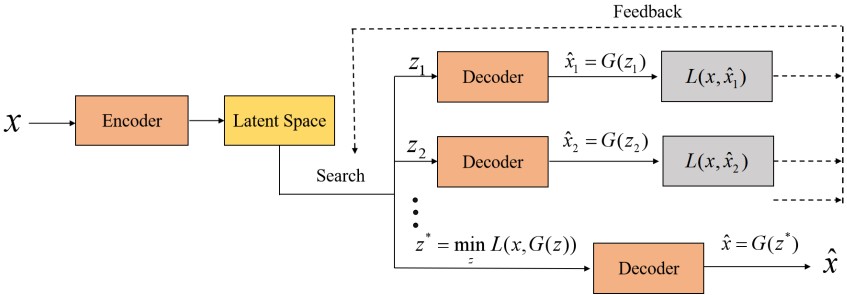

Figure 1: Framework of the encoder assisted search process in the inference stage. The encoder and decoder are pretrained. The search process is implemented in the latent space formed by the encoder and starts at the point given by the encoder. The discrepancy between the input and the reconstruction can act as the feedback from the visual space to guide the search.

performance, which brings a high computation cost. That's because the GD minimization is sensitive to the initial points, which is given randomly in Defense-GAN.

Our approach takes the advantages of both auto-encoder and Defense-GAN. We follow the encoder-decoder framework to build the reconstruction model but also include a search process to find the best reconstruction. The encoder-decoder framework can help to form a self-organized latent space and preserves the neighborhood relations (Xu, 2019), which can provide a fair start point and a more conducive environment for the search process. However, the traditional auto-encoder is trained to form an identical mapping from the input to output instead of minimizing the distance between the distribution of natural and generated images. Consequently, auto-encoder may find the closest image to a given perturbed image but out of the distribution of real data, which is still hard for a classifier to give a right prediction. To solve this problem, we add a discriminator to distinguish between natural images and generated images and force the decoder to model the distribution of natural data. The training for the reformer network is simple and requires no adversarial samples. Experimental results show that the decoder can always generate natural samples when the encoder can provide a good start point and a self-organized latent space for the search process. Besides, the encoder-decoder framework can also act as an effective detector (Meng & Chen, 2017). In the inference stage, when there comes a test sample, we can first detect if the sample is a malicious image or not. If not, we simply reconstruct the input image to eliminate missing noise. If true, we start the search process in the latent space, which is implemented by GD minimization, to find the closest natural reconstruction to the given image. Compared with Defense-GAN, we have two advantages: (1) The detecting enables us to discriminate natural samples and adversarial samples effectively, thus we can bypass the searching stage on natural samples, which saves much time; (2) For malicious samples, the search process starts at a good initial, thus we need no random restarts and only a few iterations. Compared with auto-encoder, we have a strong defense against white-box attacks due to the searching process. Experiments show that our method is robust against different attacks and can reach comparable even better performance with Defense-GAN while using much fewer computations.

## 2    RELATED WORK

### ADVERSARIAL ATTACKS

Various attack models have been studied to generate adversarial examples. Szegedy et al. first introduced adversarial examples against neural networks (Szegedy et al., 2013) and generated adversarial perturbations using an L-BFGS method to solve an optimization problem. For better efficiency, Goodfellow et al. proposed Fast Gradient Sign Method (FGSM) (Goodfellow et al., 2014b), an efficient single-step attack where the perturbation is updated along with the sign of the gradient of the loss w.r.t the input images. Later, Basic Iterative Method (BIM) (Kurakin et al., 2016b) is proposed as an extension of FGSM by running a finer optimization for multiple iterations and clipped pixel values in each iteration. After that, Moosavi-Dexfooli et al. proposed DeepFool (Moosavi-Dezfooli et al., 2016) to find the closest distance from the normal input to the decision boundary of adversar-

ial samples. More recently, Carli and Wagner (Carlini & Wagner, 2016) launched a powerful attack to defeat defensive distillation (Papernot et al., 2015), which generates adversarial examples with small perturbations. It is also possible to use generative models to generate universal adversarial perturbations (Reddy Mopuri et al., 2018).

DEFENSE METHODS

In recent years, a variety of techniques have been proposed to defend against adversarial attacks. Adversarial training (Kurakin et al., 2016a; Goodfellow et al., 2014b; He et al., 2017; Madry et al., 2017b) is one of the most popular ways which trains the classifier on the augmented training set with adversarial samples. It improves the accuracy on adversarial samples but can not generalize well to new attacks (He et al., 2017). Meanwhile, it is time-consuming to generate adversarial examples and retrain the classifier. Gradient masking method (Papernot et al., 2015) has also been considered to mask or reduce the gradients in magnitude when training the classifier, but it has been shown to give a false sense of security in defending against attacks (Athalye et al., 2018). The two popular methods include retraining the classifier, while the classifier may not be allowed to be modified in the physical world.

Another way is to pre-process adversarial samples before feeding them into classifiers. Some previous work resort to traditional denoising methods (Das et al., 2017; Dziugaite et al., 2016; Osadchy et al., 2017) to purify the adversarial examples. The main drawback of these methods is that they can only fix small perturbations and may cause information loss. Denoising auto-encoder (Gu & Rigazio, 2014) is firstly proposed to defend against adversarial attacks by training an auto-encoder to map the corrupted images to clean ones. As discussed in (Samangouei et al., 2018), the stacked network is still easy to be attacked. MagNet ensembles many auto-encoders and applies a two-step defense mechanism where the hard adversarial attacks can be detected and soft adversarial attacks can be reformed. However, MagNet still suffers from white-box attacks. Different from auto-encoder, Defense-GAN (Samangouei et al., 2018) is proposed to leverage the representative power of GAN to diminish adversarial perturbations. At the inference stage, they implement a gradient descent minimization to find the best code corresponding to the closest reconstruction. Since the gradient can not pass the searching process, Defense-Gan can strongly defense against white-box attacks while the searching process is time-consuming.

## 3 APPROACH

### 3.1 THREAT MODEL

We consider the following types of adversarial attacks, according to the level of how much information is available to the attackers:

- Black-box attacks: no details about the classifier and defense mechanism are accessible. A substitute network is trained to mimic the classifier and then is used to generate adversarial examples.
- White-box attacks: complete access to all the information about the classifier and the defense strategy.
- Gray-box attacks: complete access to the classifier but no access to defense strategy.

### 3.2 METHOD OVERVIEW

Let $f_\theta(x) : \mathbb{R}^d \to \mathbb{N}$ be a classifier parameterized by $\theta$, where $\mathbb{R}^d$ is the image space and $\mathbb{N}$ is the set of natural numbers denoting class labels. Given a clean image $x$, an adversary can perturb the clean image with small perturbation $\eta$ but confuse $f$:

$$f_\theta(x + \eta) \neq f_\theta(x), \|\eta\| < \epsilon_{attack} \tag{1}$$

where $\|\cdot\|$ is a measurement and $\epsilon_{attack}$ is the perturbation scale which sets the maximum perturbation allowed for each pixel and often set to a small number to get almost imperceptible difference between $x_{adv} = x + \eta$ and $x$.

We want to improve robustness by learning a transformation $T(.)$, such that the predicted label of the transformed image does not change when the image corrupted with perturbations, i.e.,

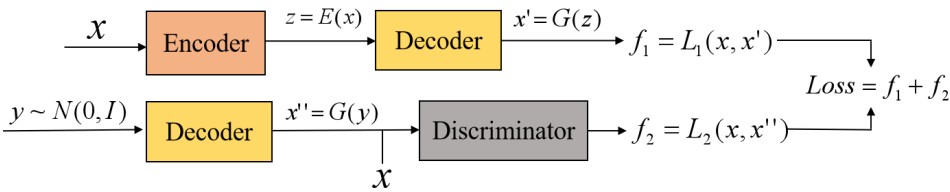

Figure 2: Architecture of AE-GAN

$f_\theta(T(x_{adv})) = f_\theta(T(x))$. The basic idea of our method is to purify input images by searching the closest reconstruction without perturbations within little computation cost. To achieve this, we build a reconstruction network based on the encoder-decoder framework, which is denoted as AE-GAN$_{+r}$ and will be described in detail in Section 3.3. After training, the encoder of AE-GAN$_{+r}$ can provide a good representation for most inputs, and the decoder can always generate an image on the manifold of natural samples during the search.

In the inference time, when given an input, we first detect if the sample is adversarial or not. If not, we simply reconstruct the input image by AE-GAN$_{+r}$ to diminish missing noises. If true, we start a search process in the latent coding space to find the best code, which corresponds to the closest reconstruction without perturbations. The search process starts at the output of the encoder and implemented by GD minimization, which will be described in Section 3.4. Formally, we denote this encoder assisted search process as AE-GAN$_{+rs}$.

### 3.3 ARCHITECTURE OF AE-GAN

An auto-encoder (AE) is composed of two components: the encoder $F$ and the decoder $G$. Let $\mathcal{X}$ be the visual space and $\mathcal{Z}$ be the latent space. The mapping $F : \mathcal{X} \to \mathcal{Z}$ performs image abstraction, which encode the image $x \in \mathcal{X}$ into a code $z \in \mathcal{Z}$. The mapping $\mathcal{Z} \to \mathcal{X}$ performs image generation, which generates image $x \in \mathcal{X}$ from the output of encoder $z \in \mathcal{Z}$. Formally, we denote the distribution of training images as $p_{data}$. The loss function of AE is defined as:

$$\mathbb{E}_{x \sim p_{data}} \|G(F(x)) - x\|_2^2 \tag{2}$$

As shown in Equation 2, traditional AE minimizes the euclidean distance between input and output. Thus, it is trained to form an identical mapping from the input to output instead of minimizing the distance between the distribution of natural images $p_{data}$ and generated images $p_g$. Consequently, when we search for the best code $z$ in the latent space, AE may find a close image $G(z)$ but out of $P_{data}$, as shown in Figure 5 in Appendix E, which is a meaningless purification because the accuracy of classifier on these images is not good. To solve this problem, we adapt the auto-encoder by introducing a discriminator and the min-max loss in GANs (Goodfellow et al., 2014a), as shown in Figure 2. We denote the adapted AE as AE-GAN. GANs consists of two neural networks, a generator $G$ and a discriminator $D$. $G$ maps a low-dimensional latent space $\mathcal{Z}$ to the high dimensional sample space $\mathcal{X}$. $D$ is a binary neural network classifier. While $G$ learns to generate samples that similar to the real data $x$, $D$ learns to distinguish between "real" samples $x$ and "fake" samples $G(z)$. $D$ and $G$ are trained in an alternating fashion to minimize the following min-max loss (Goodfellow et al., 2014a):

$$\min_G \max_D V(D, G) = \mathbb{E}_{x \sim p_{data}}[\log D(x)] + \mathbb{E}_{z \sim p_z}[\log(1 - D(G(z))] \tag{3}$$

To preserve the advantages of AE, we still keep the mean square error to drive the abstraction mapping and self-organizing in the latent space. Meanwhile, we add a discriminator learning to distinguish between the samples generated by the decoder and real samples. Thus, the objective of AE-GAN is to minimize the following min-max loss:

$$\min_{F,G} \max_D V(D, F, G) = \mathbb{E}_{x \sim p_{data}}[log(D(x))] + \mathbb{E}_{z \sim p_z}[log(1 - D(G(z)))] +$$
$$\mathbb{E}_{x \sim p_{data}} \|G(F(x)) - x\|_2^2 \tag{4}$$

As the start point of the search process, the output of the encoder is expected to be close to the optimum code and thus help the search to converge quickly. Since AE-GAN is trained with legitimate

images, the encoder $F$ may suffer from the covariate shift when the input is corrupted with large perturbations. To solve this, we perturb the clean images with random Gaussian noise before feeding them into the encoder when calculating the reconstruction error. Adding noise in the input layer acts as a regularization (Bishop, 1995), thus we denote the method as noise regularization $(+r)$ in this paper. The final objective function of AE-GAN$_{+r}$ can be written as Equation 5.

$$
\min_{F,G} \max_{D} V(D, F, G) = \mathbb{E}_{x \sim p_{data}}[log(D(x))] + \mathbb{E}_{z \sim p_z}[log(1 - D(G(z)))] +
$$
$$
\mathbb{E}_{x \sim p_{data}} \|G(F(x + \eta)) - x\|_2^2 ,
\tag{5}
$$

where $\eta \sim N(0, \sigma)$, $\sigma$ follows the uniform distribution on a given region.

Bidirectional Generative Adversarial Networks (BiGANs) (Donahue et al., 2016) also includes encoder-decoder mapping. In addition to the generator from the standard GAN framework, BiGAN includes an encoder to map the input data into the latent representation. The discriminator in BiGAN discriminates jointly data and latent code, i.e., $(x, E(x))$ versus $(G(z), z)$). However, experimental results show that BiGAN has poor reconstruction performance. When BiGAN equipped with the searching process, the encoder in BiGAN can not provide a good initial point for the decoder.

### 3.4 ENCODER ASSISTED SEARCH PROCESS

Although AE-GAN$_{+r}$ has been trained with noise regularization, the added noise is simple while the adversarial noise is carefully crafted, the distribution between them is different. Moreover, when the input $x^*$ is seriously corrupted, e.g., a large region of the image is masked, the code derived from encoder $F(x^*)$ can hardly represent the original image $x$. Therefore, a correction for $F(x^*)$ is required when the input pattern is significantly different from the training data. Meanwhile, the end-to-end encoder-decoder framework can not perform well against white-box attacks (Samangouei et al., 2018). Therefore, we include a search (denoted as $+s$) process in the latent space in the inference stage.

As shown in Figure 1, the core idea is that for every input sample $x$, AE-GAN$_{+s}$ will process a search process in the latent space to find the best code for an input, which can lead the decoder to generate the closest reconstruction. For a given input $x$, we define the objective function $J$ of the search process as:

$$
J(z) = L(x, G(z)), \text{for } z \in \mathcal{Z}
\tag{6}
$$

where $G(.)$ denotes the decoder, $L(.)$ is a measurement of the discrepancy between $x$ and $G(z)$. We aims to find the optimum code $z^*$ such that

$$
z^* = \arg\min_{z \in \mathcal{Z}} J(z)
\tag{7}
$$

Since our latent space is designed to be continuous, thus the direction and step size for the search process can be calculated by Gradient Decent (GD) method. Therefore, the search process is converted to a GD minimization to minimize the discrepancy between the input and the reconstruction. Here we simply use euclidean distance on the pixel level to measure the discrepancy. A more detailed description of the searching process is stated in Appendix C.

In our framework, the encoder forms a self-organized latent space that preserves neighborhood relations and provides a good start point. Both of them can assist the search process in convergence. In Defense-GAN (Samangouei et al., 2018), the latent space follows a single Gaussian distribution while the gradient descent method is sensitive to initial values. Therefore, it requires multiple random restart points and considerable iterations to guarantee the performance, which brings a high computation cost.

## 4 EXPERIMENTS

### 4.1 SETTINGS

In this section, we carry out our experiments on two image datasets: the MNIST (LeCun et al., 1998) and F-MNIST (Xiao et al., 2017) dataset. Both of them contain $60,000$ training images and $10,000$ testing images. We split the original training set by 20:1 for training and validation, and

Table 1: Classification accuracies of classifier A using various defense strategies on the MNIST (top) and F-MNIST (bottom) dataset, under FGSM ($\epsilon = 0.3$), PGD ($\epsilon = 0.3$) and CW (with $L_2$ norm) white-box attacks, gray-box attacks, and FGSM black-box attacks. Here the corresponding iterations of GD minimization are: 15 for our method, 60 for BiGAN, and 200 for Defense-GAN. (Adv.Tr.: adversarial training with PGD ($\epsilon = 0.3$))

| Attack | Method | None | Our | Adv. Tr. | MagNet | BiGAN$_{+s}$ | Defense-Gan |
|--------|--------|------|-----|----------|--------|--------------|-------------|
| White | FGSM | 0.144 | **0.984** | 0.949 | 0.293 | 0.912 | 0.981 |
|  | PGD | 0.007 | 0.982 | 0.920 | 0.012 | 0.924 | **0.989** |
|  | CW | 0.008 | 0.979 | 0.772 | 0.013 | 0.905 | **0.981** |
| Gray | FGSM | 0.144 | **0.882** | - | 0.561 | 0.731 | 0.880 |
|  | PGD | 0.007 | 0.910 | - | 0.673 | 0.739 | **0.913** |
|  | CW | 0.008 | 0.915 | - | 0.881 | 0.805 | **0.918** |
| Black | A/B | 0.701 | 0.935 | **0.971** | 0.464 | 0.701 | 0.928 |
|  | A/C | 0.813 | 0.933 | **0.970** | 0.590 | 0.632 | 0.922 |
| Attack | Method | None | Our | Adv. Tr. | MagNet | BiGAN$_{+s}$ | Defense-Gan |
| White | FGSM | 0.073 | 0.804 | 0.739 | 0.199 | 0.613 | **0.814** |
|  | PGD | 0.028 | 0.816 | 0.717 | 0.071 | 0.653 | **0.852** |
|  | CW | 0.062 | 0.767 | 0.308 | 0.084 | 0.601 | **0.810** |
| Gray | FGSM | 0.073 | **0.607** | - | 0.304 | 0.570 | 0.389 |
|  | PGD | 0.028 | **0.736** | - | 0.427 | 0.605 | 0.491 |
|  | CW | 0.062 | **0.719** | - | 0.646 | 0.608 | 0.435 |
| Black | A/B | 0.227 | 0.603 | **0.784** | 0.308 | 0.398 | 0.602 |
|  | A/C | 0.295 | 0.507 | **0.777** | 0.267 | 0.348 | 0.461 |

keep the original testing set as our testing set. Besides, we randomly choose 2000 samples from the testing set as a development dataset, shortly dev-set, on which we conduct some exploratory experiments. For the architecture and training of the target classifier A, we follow the settings in Defense-GAN(Samangouei et al., 2018) and the details are described in Appendix B. After training, we get an accuracy of 99.0% and 91.8% on MNIST and F-MNIST respectively, which near state of the art. For the architecture and training details of AE-GAN, we adopt the settings in Defense-GAN (Samangouei et al., 2018) and restate them in Appendix B with a detailed description.

Specifically, we consider the white-box and gray-box attacks by FGSM (Goodfellow et al., 2014b), PGD (Madry et al., 2017a) and Carlini-Wagner (CW) attack with $\ell_2$ norm (Carlini & Wagner, 2016), and black-box attacks by FGSM. The training procedure of the substitute model in black-box attacks is the same as the setting in (Papernot et al., 2017). We split the testing set into a small hold-out set of 150 samples for training the substitute model and the remaining 9850 samples for testing different defense strategies. For all the attack methods, we use the implementation in Cleverhans (Papernot et al., 2018), a python library to benchmark machine learning systems' vulnerability to adversarial examples, and enforce the image to remain within $[0, 1]^{H \times W}$ by clipping.

## 4.2 RESULTS ON ADVERSARIAL ATTACKS

We compare our method to PGD adversarial training (Madry et al., 2017b), MagNet (Meng & Chen, 2017), Defense-Gan (Samangouei et al., 2018), BiGAN (Donahue et al., 2016) under the FGSM, PGD and CW (with $l_2$ norm) white-box and gray-box attacks, as well as the FGSM black-box attacks. Our reported method is AE-GAN$_{+rs}$, i.e., AE-GAN with noise regularization and search process, where the number of iterations $L = 15$. We also assist BiGAN with the search process for fair comparisons, which is denoted as BiGAN$_{+s}$. Reported Defense-GAN follows the original set-

tings where iterations $L = 200$ and restart points $R = 10$. All experimental results are summarized in Table 1.

From Table 1, we can observe that adversarial training successfully defends against the FGSM and PGD attack, but can not generalize to CW attacks. Because it needs adversarial examples to retrain the classifier. As discussed in Section 3.3, on small scale perturbations like CW the performance of MagNet and AE-GAN$_{+rs}$ is close, because the code derived from the encoder is robust to small perturbations. However, on large scale perturbations like FGSM ($\epsilon = 0.3$) and PGD ($\epsilon = 0.3$), the reported AE-GAN$_{+rs}$ can achieve about $20\% \sim 30\%$ improvements than MagNet.

As shown in Table 1, adversarial training and MagNet can only perform well under specific situations, while Defense-GAN and our method can defense against various attacks. Attack-agnostic is important for a defense strategy because we have no information about the attacker in the physical world. While our method reaches comparable performance with Defense-GAN under white-box and black-box attacks, we significantly reduce the computation complexity to $15/2000$ of Defense-GAN. Another observation is that when the performance both decreases on the F-MNIST dataset due to the large perturbations, our method performs much better than Defense-GAN on gray-box attacks. The possible reason is that images in F-MNIST are more complex and require more details, which raises the difficulty of searching the best code in the latent space from random points. For white-box attacks, the performance of our method has a slight drop from Defense-GAN. However, it has been studied in (Samangouei et al., 2018) that more iterations can increase the robustness for Defense-GAN against GD-based white-box attacks. When we decline $L$ from 200 to 15 for Defense-GAN, the performance drops by $8\%$ decline. We can also observe that with fewer iterations, AE-GAN$_{+rs}$ still performs much better than BiGAN$_{+s}$ under various attacks. Though BiGAN$_{+s}$ also includes an encoder-decoder mapping, the encoder of BiGAN$_{+s}$ can not provide a good start point for the searching process. This is due to the poor reconstruction problem of BiGAN, which will be further studied in Section 4.5.

Table 1 also shows that the performance of defenses against gray-box and black-box attacks on the F-MNIST dataset is noticeably lower than on MNIST. This is due to the large perturbations generated by the FGSM attack with $\epsilon = 0.3$. It is more difficult to find an accurate reconstruction for the seriously perturbed image, as shown in Figure 10 in Appendix H.

### 4.3 TRADE-OFF BETWEEN COMPUTATION AND PERFORMANCE

We further investigate the trade-off between computation cost and performance in Defense-GAN and our method. Table 2 shows the robustness of Defense-GAN and AE-GAN$_{+rs}$ when changing the number of restart points $R$ and iterations $L$. It can be seen that with fewer restart points, the performance of Defense-GAN decreases a lot after a certain $L$ value, which means that the effect of varying $R$ is extremely pronounced. This is due to the non-convex nature of MSE and increasing $R$ enables Defense-GAN to sample different local minimum (Samangouei et al., 2018). However, in our framework, taking the output of the encoder as the initial start point can enable the search process to reach a reasonable local minimum in a few iterations. When we decrease $L$ from 60 to 10, there is $6\% - 10\%$ drop for the accuracy of Defense-GAN, while there is little change in the performance of AE-GAN$_{+rs}$. Therefore, compared with Defense-GAN, our method requires less computation than Defense-GAN when they reach comparable performance. The performance of our is robust to the computation.

### 4.4 DETECTION

A trained AE-GAN$_{+r}$ can well reconstruct natural images without searching. From this observation, we propose to introduce the detection mechanism to bypass the searching process for clean images. In AE-GAN$_{+r}$, the decoder is trained to produce images that resemble the legitimate data. Thus, the adversarial examples usually have a high reconstruction error. Therefore, we can choose the mean square error (MSE) between input and reconstruction as our indicator to decide whether or not the image is adversarial. The setting for the threshold $\eta$ is important for the detection. A larger threshold can tolerant more perturbations and enhance efficiency but reduce security.

To investigate the performance of detection, we compute the reconstruction error for every image in the testing dataset, and their corresponding adversarially perturbed images under different strengths

Table 2: Classification accuracy of classifier A using different defense strategy on test set of M-NIST (left) and F-MNIST (right), under FGSM ($\epsilon = 0.3$) attack, with different GD iterations $L = 10, 60, 200$ and various numbers of restart points $R = 1, 10$.

| Attack | Method | $R = 1$ | | $R = 10$ | |
|--------|--------|---------|---------|----------|---------|
| | | $L = 10$ | $L = 60$ | $L = 10$ | $L = 60$ |
| Clean | Our | **0.982\0.837** | **0.985\0.841** | - | - |
| | Defense-Gan | 0.545\0.405 | 0.693 \0.530 | 0.909 \ 0.696 | 0.967\ 0.788 |
| White | Our | **0.984\0.799** | **0.985\0.804** | - | - |
| | Defense-Gan | 0.537\ 0.403 | 0.688\0.527 | 0.905\0.697 | 0.964\0.787 |
| Gray | Our | **0.880 \0.604** | **0.884\0.595** | - | - |
| | Defense-Gan | 0.398\0.296 | 0.570 \0.325 | 0.769\ 0.441 | 0.863\0.432 |

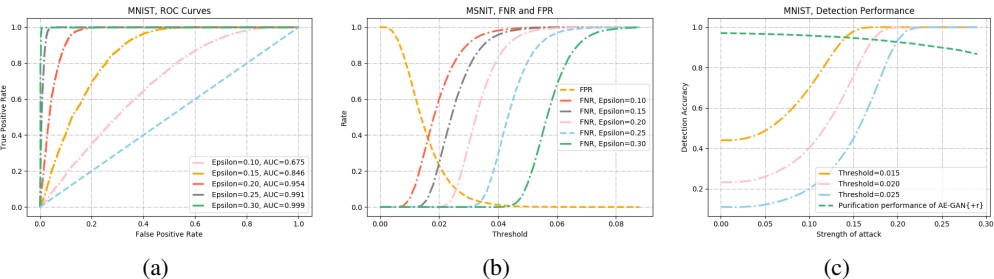

(a)  (b)  (c)

Figure 3: Detections on MNIST. Left: ROC curves when using AE-GAN$_{+r}$ to detect FGSM attacks with various $\epsilon$. Middle: The False Negetive Rate and False Positive Rate when using different thresholds to detect FGSM attacks with various $\epsilon$. Right: Detection performance on different strength of FGSM attacks on MNIST dataset and the purification performance of AE-GAN$_{+r}$.

of FGSM attacks. Figure 3 (a) shows the Receiver Operating Characteristic (ROC) curves as well as the Area Under the Curve (AUC) metric, which indicates that the detection is effective especially for large perturbations. When the threshold $\eta$ increases, we calculate the False Negative Rate (FNR,the percentage of missed diagnosis) and False Positive Rate (FPR, the percentage of false alarms) and visualize how FNR and FPR changes with $\eta$ under different strengths of attacks in Figure 3 (b). The results show that a smaller threshold corresponds to lower FNR but higher FPR, which makes the detection safer but may cause false alarms. Meanwhile, when the perturbation is small, it is hard to detect attacks at both low FNR and FPR, but AE-GAN$_{+r}$ is good at purifying small perturbations, which can make up this shortcuts, as shown in Figure 3 (c). We also test the detection performance when using $L_1$ loss as the indicator, which is shown to be more effective. It can reach both small FNR and FPR even under small perturbations. For more details, please refer to Appendix F.

In summary, for small perturbations which pass the detection, AE-GAN$_{+r}$ can successfully eliminate them. For larger perturbations that are detected as adversarial, they will trigger the searching stage and be purified by AE-GAN$_{+rs}$ for achieving higher accuracy.

## 4.5 ABLATION STUDY

To investigate the effect of each module in our method, we compare the performance of different models under the FGSM gray-box attack and half-masked attack (half of the input image is blocked, as shown in Figure 5 (c) in Appendix E. Table 3 shows the defense performance of the different models on different image corruptions visualized in Figure 5, where the number of iterations $L$ is set to be 20 for models with search process.

Firstly, by comparing the performance of AE-GAN and AE-GAN$_{+r}$, we can observe that noise regularization has an expected effect of forming a more accurate latent code for the corrupted input. Secondly, the accuracies of AE$_{+s}$ and AE-GAN$_{+s}$ as well as the qualitative results in Figure 5 (see

Table 3: Classification accuracies of classifier A using various defense strategies on the MNIST dev-set, under FGSM ($\epsilon = 0.3$), CW (with $l_2$ norm) gray-box attack and Half-Masked (H.M.) attack. (A.G.: AE-GAN; B.G.: BiGAN; +s: includes search process; +r: with noise regularization.)

| D | AE | AE$_{+s}$ | B.G. | B.G.$_{+s}$ | A.G. | A.G.$_{+s}$ | A.G.$_{+r}$ | A.G.$_{+rs}$ |
|------|-------|-------|-------|-------|-------|-------|-------|-------|
| FGSM | 0.515 | 0.678 | 0.428 | 0.688 | 0.502 | 0.781 | 0.806 | **0.832** |
| CW | 0.861 | 0.852 | 0.477 | 0.749 | 0.868 | 0.887 | 0.870 | **0.889** |
| H.M | 0.552 | 0.672 | 0.408 | 0.590 | 0.494 | **0.807** | 0.570 | 0.776 |

Appendix E) reveals that the introduced discriminator and adversarial loss improve the generative ability of the decoder. Thirdly, we also look at the consistency between input and output of BiGAN. As illustrated in Table 3, the defense performance of BiGAN is not good which indicates that the output of the encoder is sometimes inconsistent with the original images, thus the start point can not assist the searching process in BiGAN$_{+s}$ to reach a local minimum within a certain number of steps.

## 5 CONCLUSION

We proposed AE-GAN$_{+rs}$, a framework to effectively defend against various adversarial attacks. Our method takes the advantages of both auto-encoder and GANs and avoids their drawbacks. We build a reconstruction network following the encoder-decoder framework, as well as introduce the search process in the latent space to find the closet reconstruction without perturbations to the given image. Compared with auto-encoder based methods, we can strongly defend against white-box attacks. Compared with GANs method, which requires multiple random restart points and considerable iterations to purify a sample, the encoder in our framework can help the search process converge quickly. We empirically show that Defense-GAN and our method can effectively defend against various attacks, while other methods have many shortcomings on at least one type of attack. Meanwhile, we can achieve comparable even better performance than Defense-GAN with a significant reduction of computations.

The success of AE-GAN relies on both the representative power of the encoder and the generative power of the generator. However, as stated in Defense-GAN (Samangouei et al., 2018), training GANs is still a challenging task and an active area of research. Meanwhile, obtaining a powerful auto-encoder on a large dataset is also challenging. Besides this, we hope the proposed method provides a way to reducing the computations in searching based defense methods.

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

## A    IMPLEMENTATIONS

For adversarial training, we follow (Madry et al., 2017b) and augment the training dataset with adversarial samples generated by PGD. We use the same $\epsilon$ for training and testing and set $\epsilon = 0.3$ for MNIST and F-MNSIT, $\epsilon = 0.1$ for CelebA dataset.

For Defense-GAN, we adopt the implementation of original work (Samangouei et al., 2018).

For BiGAN, the original implementation (Donahue et al., 2016) is based on the fully-connected network. For fair comparisons, we adopt the convolutional architecture in another similar work (Dumoulin et al., 2016). We slightly adapt the architecture on CIFAR-10 (Krizhevsky et al., 2009) by changing the input channel from 3 to 1. As shown in Figure 4, the model we trained reach a fair performance on generating samples.

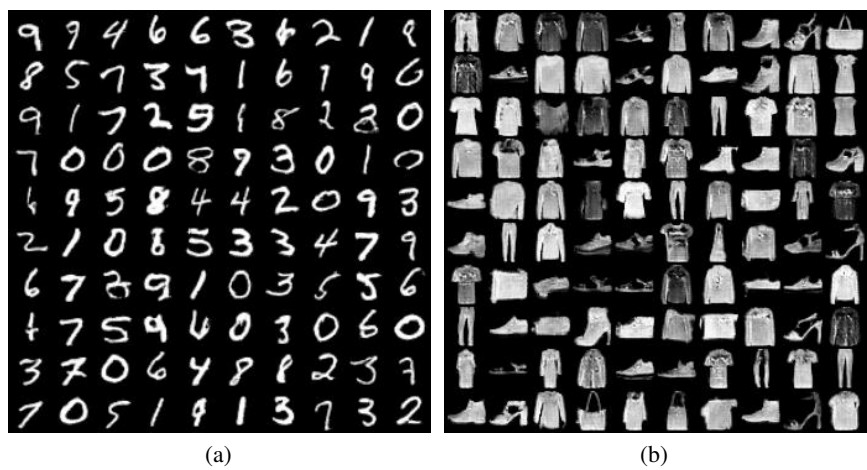

| (a) | (b) |

Figure 4: Generated samples by BiGAN trained on MNIST (left) and F-MNIST (right).

For MagNet, we follow the implementation in Defense-GAN (Samangouei et al., 2018) and restate the details in Table 6, which shares the same architecture with AE-GAN. However, we find that the latent space dimension 128 is too large for MagNet to eliminate strong perturbations. Because the details of noise are also encoded and can be reconstructed. Thus, we also take the latent dimension as 20 for comparisons. The results in Table 8 show that the change can enhance the performance on the gray-box attack.

## B    NEURAL NETWORK ARCHITECTURES AND TRAINING SETTINGS

For fair comparisons, we compare different strategies through this paper on the same classifier A, and we take two substitute classifiers for black-box attacks, B for convolutional structure and C for fully connected structure. The training settings and architectures almost follow the settings in Defense-GAN(Samangouei et al., 2018). For clarification, here we restate their architectures and training settings in Table 5 and Table 7 respectively.

The architecture of our model and MagNet is described in Table 6, which make a slight modification on the MagNet structure given in (Samangouei et al., 2018)

Table 4: Training parameters for AE-GAN and MagNet

| Parameters | MNIST, F-MNIST | CelebA |
|---|---|---|
| Epochs | 60 | 100 |
| Learning Rate | 0.0002 | 0.0002 |
| Optimization Method | Adam | Adam |
| Batch Size | 100 | 64 |

Table 5: Neural architectures used for classifiers and substitute models

| A | B | C |
|---|---|---|
| Conv(64, $5 \times 5$, 1) | Conv(64, $8 \times 8$, 2) | |
| ReLU | ReLU | FC(200) |
| Conv(64, $5 \times 5$, 2) | Conv(128, $6 \times 6$, 2) | ReLU |
| ReLU | ReLU | Dropout(0.5) |
| Dropout(0.25) | Conv(128, $5 \times 5$, 1) | FC(200) |
| FC(128) | ReLU | ReLU |
| ReLU | Drpupout(0.5) | Dropout(0.5) |
| Drpupout(0.5) | FC(10)+Softmax | FC(10)+Softmax |
| FC(10)+Softmax | | |

Table 6: Neural architectures used for AE-GAN and MagNet

| Encoder | Decoder | Discriminator |
|---|---|---|
| Conv(1, 64, $5 \times 5$, 2) | FC(1024) & ReLU | Conv(1, 64, $5 \times 5$, 2) |
| LeakyReLU(0.2) | ConvT(256, 128, $5 \times 5$, 2) | LeakyReLU(0.2) |
| Conv(64, 128, $5 \times 5$, 2) | BatchNorm & ReLU | Conv(64, 128, $5 \times 5$, 2) |
| BatchNorm & LeakyReLU | ConvT(128, 64, $5 \times 5$, 2) | BatchNorm & LeakyReLU |
| Conv(128, 256, $5 \times 5$, 2) | BatchNorm & ReLU | Conv(128, 256, $5 \times 5$, 2) |
| BatchNorm & LeakyReLU | ConvT(64, 1, $5 \times 5$, 2) | BatchNorm & LeakyReLU |
| FC(128)+tanh | Sigmoid | FC(1)+Sigmoid |

Table 7: Training parameters for classifiers

| Parameters | MNIST, F-MNIST |
|---|---|
| Epochs | 10 |
| Learning Rate | 0.001 |
| Optimization Method | Adam |
| Batch Size | 100 |

## C DETAILED DESCRIPTION OF THE TRAINING AND INFERENCE STAGE.

Algorithm 1 describes the training process of AE-GAN$_{+r}$. Algorithm 2 describes the search process of AE-GAN$_{+rs}$ in the inference stage.

---
**Algorithm 1 Training process of AE-GAN$_{+r}$**

---
**Notations:** $F$: encoder; $G$: decoder; $D$: discriminator; $\eta$: the bound of random noise

    **while** $epoch < iteration$ **do**

        Sample minibatch of $m$ noise samples $\{z^{(1)}, z^{(2)}, \ldots, z^{(m)}\}$ from noise prior $p(z)$

        Sample minibatch of $m$ samples $\{x^{(1)}, x^{(2)}, \ldots, x^{(m)}\}$ from training set

        Update the discriminator by ascending its stochastic gradient

$$\nabla_{\theta_d} \frac{1}{m} \sum_{i=1}^{m} \left[\log D\left(x^{(i)}\right) + \log\left(1 - D\left(G\left(z^{(i)}\right)\right)\right)\right]$$

        Sample $\sigma$ from uniform distribution $U(0, \eta)$

        Sample minibatch of $m$ noise samples $\{\epsilon^{(1)}, \epsilon^{(2)}, \ldots, \epsilon^{(m)}\}$ from $N(0, \eta)$

        Update the decoder by descending its stochastic gradient

$$\nabla_{\theta_g} \frac{1}{m} \sum_{i=1}^{m} [\log\left(1 - D\left(G\left(z^{(i)}\right)\right)\right) + (F(z^{(i)}) - (x^{(i)} + \epsilon^{(i)}))^2]$$

        Update the encoder by descending its stochastic gradient.

$$\nabla_{\theta_e} \frac{1}{m} \sum_{i=1}^{m} (F(z^{(i)}) - (x^{(i)} + \epsilon^{(i)}))^2$$

    **end while**

---

---

**Algorithm 2 Search process of AE-GAN$_{+rs}$ in inference stage**

---

**Input:** input image $x$, trained encoder $F$, trained decoder $G$, number of steps $L$, initial step size $\eta$
**Output:** Optimal reconstruction $G(z_k)$ of $x$
   initiate $z_0 = F(x)$
   **while** $k < L$ **do**
      $z_k = z_{k-1} + \eta \nabla_z(L_2(G(z_k) - x))$
      **if** k mod 5 == 0 **then**
         $\eta = \eta \times \frac{1}{2}$
      **end if**
   **end while**
   **return** $G(z_k)$

---

# D   ADDITIONAL RESULTS ON VARIOUS LATENT DIMENSIONS

Table 8: Classification accuracy of A using MagNet and AE-GAN on MNIST with various latent space dimensions. D: the dimension of latent space (the number of iterations is set to 15).

| Attack | Attack | None | D=20 | | D=128 | |
|---|---|---|---|---|---|---|
| | | | MagNet | AE-GAN$_{+rs}$ | MagNet | AE-GAN$_{+rs}$ |
| White | FGSM | 0.144 | 0.298 | 0.972 | 0.293 | 0.984 |
| | PGD | 0.007 | 0.012 | 0.975 | 0.006 | 0.982 |
| | CW | 0.008 | 0.013 | 0.970 | 0.008 | 0.979 |
| Gray | FGSM | 0.144 | 0.561 | 0.875 | 0.316 | 0.882 |
| | PGD | 0.007 | 0.673 | 0.911 | 0.145 | 0.910 |
| | CW | 0.008 | 0.881 | 0.907 | 0.345 | 0.915 |

# E   QUALITATIVE COMPARISONS BETWEEN AE$_{+s}$ AND AE-GAN$_{+s}$

Figure 5 shows the purification performance of AE$_{+s}$ and AE-GAN$_{+s}$ on corrupted images. We can easily see that though AE$_{+s}$ incorporates a searching process, it can not find a natural reconstruction to the given corrupted image. The comparison can show the effectiveness of discriminator and the adversarial loss we introduced in AE-GAN$_{+s}$.

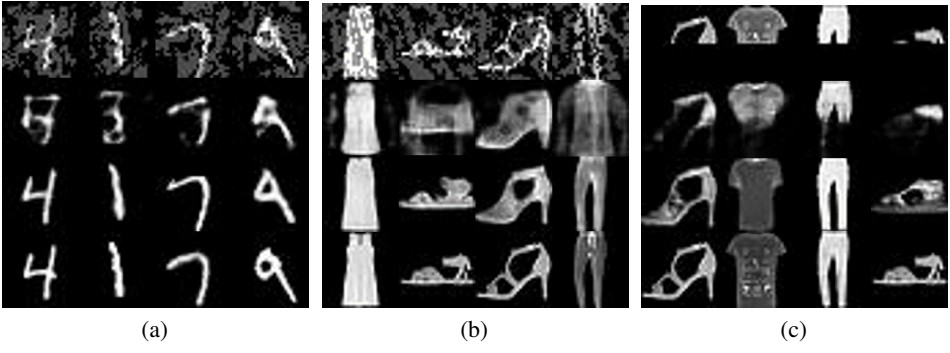

      (a)              (b)              (c)

Figure 5: Purified images on (a) MNIST and (b) F-MNIST dataset, under FGSM ($\epsilon = 0.3$) gray-box attack, as well as (c) Half-Masked attack on F-MNIST. First row: the corrupted images; Second row: purified image by AE$_{+s}$; Third row: purified image by AE-GAN$_{+s}$; Last row: ground truth.

## F    ADDITIONAL RESULTS ON DETECTION

When using $L_1$ loss as the indicator to decide whether a image is adversarial or not, we can obtain better detection performance on MNIST dataset. As shown in Figure 6, when the threshold is set to 0.06, the detection can reach both small False Negative Rate and small False Positive Rate on small perturbations, while it is hard to achieve this with MSE, as shown in Figure 3 in the main text. By comparing Figure 7 and Figure 8, we can get similar observations on F-MNIST dataset.

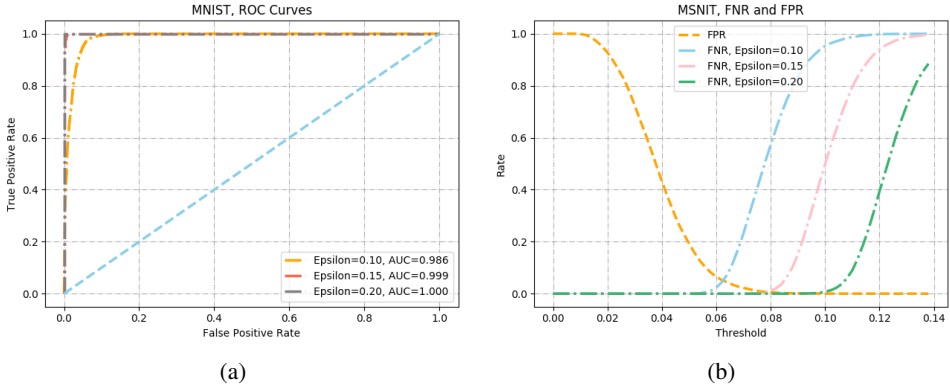

Figure 6: Detections on MNIST when using $L_1$ distance as the indicator. Left: ROC curves when using AE-GAN$_{+r}$ to detect FGSM attacks with various $\epsilon$. Right: The False Negetive Rate and False Positive Rate when using different thresholds to detect FGSM attacks with various $\epsilon$.

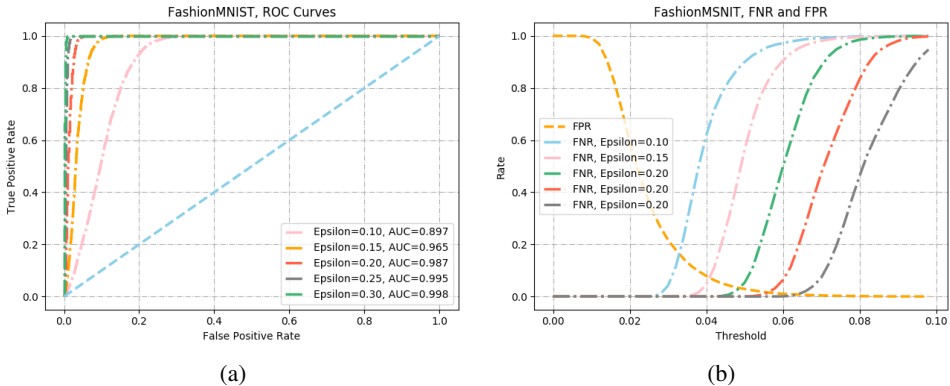

Figure 7: Detections on F-MNIST when using $L_1$ distance as the indicator. Left: ROC curves when using AE-GAN$_{+r}$ to detect FGSM attacks with various $\epsilon$. Right: The False Negative Rate and False Positive Rate when using different thresholds to detect FGSM attacks with various $\epsilon$.

## G    ADDITIONAL RESULTS ON CELEBA

To investigate the scalability of our method, we conduct experiments on CelebFaces Attributes dataset (CelebA) (Liu et al., 2015) and report the results in Table 9. CelebA contains more than 200,000 face images and we extract 162770, 10000, 10000 images for training, validation, and testing. Each image is center-cropped and resized to $64 \times 64$. Following the settings in Defense-GAN (Samangouei et al., 2018), we perform the classification of gender on the dataset and the architecture of AE-GAN also adopts the implementation in Defense-GAN for a fair comparison. The architecture of the encoder is the same as the discriminator except for modifying the dimension of the top layer. The accuracy of the trained classifier A on the training dataset of CelebA is $95.96\%$.

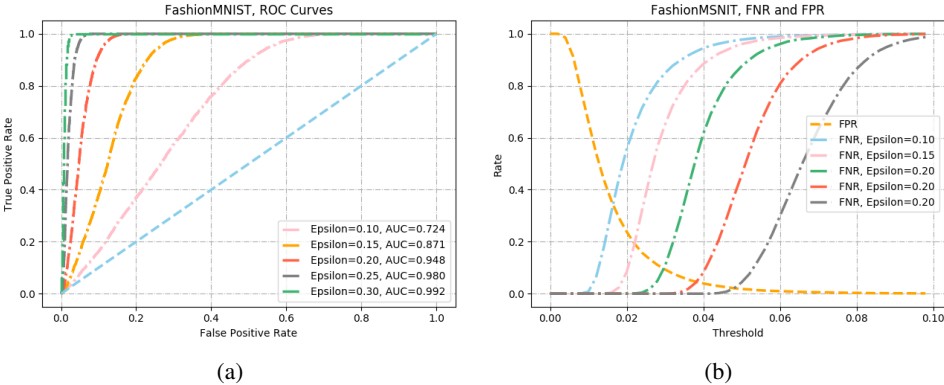

(a)                                                         (b)

Figure 8: Detections on F-MNIST when using MSE as the indicator. Left: ROC curves when using AE-GAN$_{+r}$ to detect FGSM attacks with various $\epsilon$. Right: The False Negative Rate and False Positive Rate when using different thresholds to detect FGSM attacks with various $\epsilon$.

Table 9: Classification accuracies of classifier A using various defense strategies on CelebA dataset, under FGSM ($\epsilon = 0.1$), PGD ($\epsilon = 0.1$) and CW (with $L_2$ norm) white-box, gray-box, and FGSM black-box. Here the corresponding iterations of GD minimization are: 20 for our method, 200 for Defense-GAN. (Adv.Tr.: adversarial training with PGD ($\epsilon = 0.1$))

| Attack | Method | None | Our | Adv. Tr. | MagNet | Defense-Gan |
|--------|--------|------|-----|----------|--------|-------------|
| White | FGSM | 0.0416 | **0.9364** | 0.6119 | 0.0750 | 0.8879 |
| | PGD | 0.0404 | **0.9327** | 0.6119 | 0.0510 | 0.8942 |
| | CW | 0.0404 | **0.9433** | 0.6119 | 0.0493 | 0.861 |
| Gray | FGSM | 0.0416 | 0.8872 | - | **0.8897** | 0.8559 |
| | PGD | 0.0404 | 0.9005 | - | **0.9065** | 0.8621 |
| | CW | 0.0404 | 0.9233 | - | **0.9273** | 0.867 |
| Black | A/B | 0.8325 | **0.8839** | 0.6121 | 0.8743 | 0.8517 |

## H   QUALITATIVE EXAMPLES

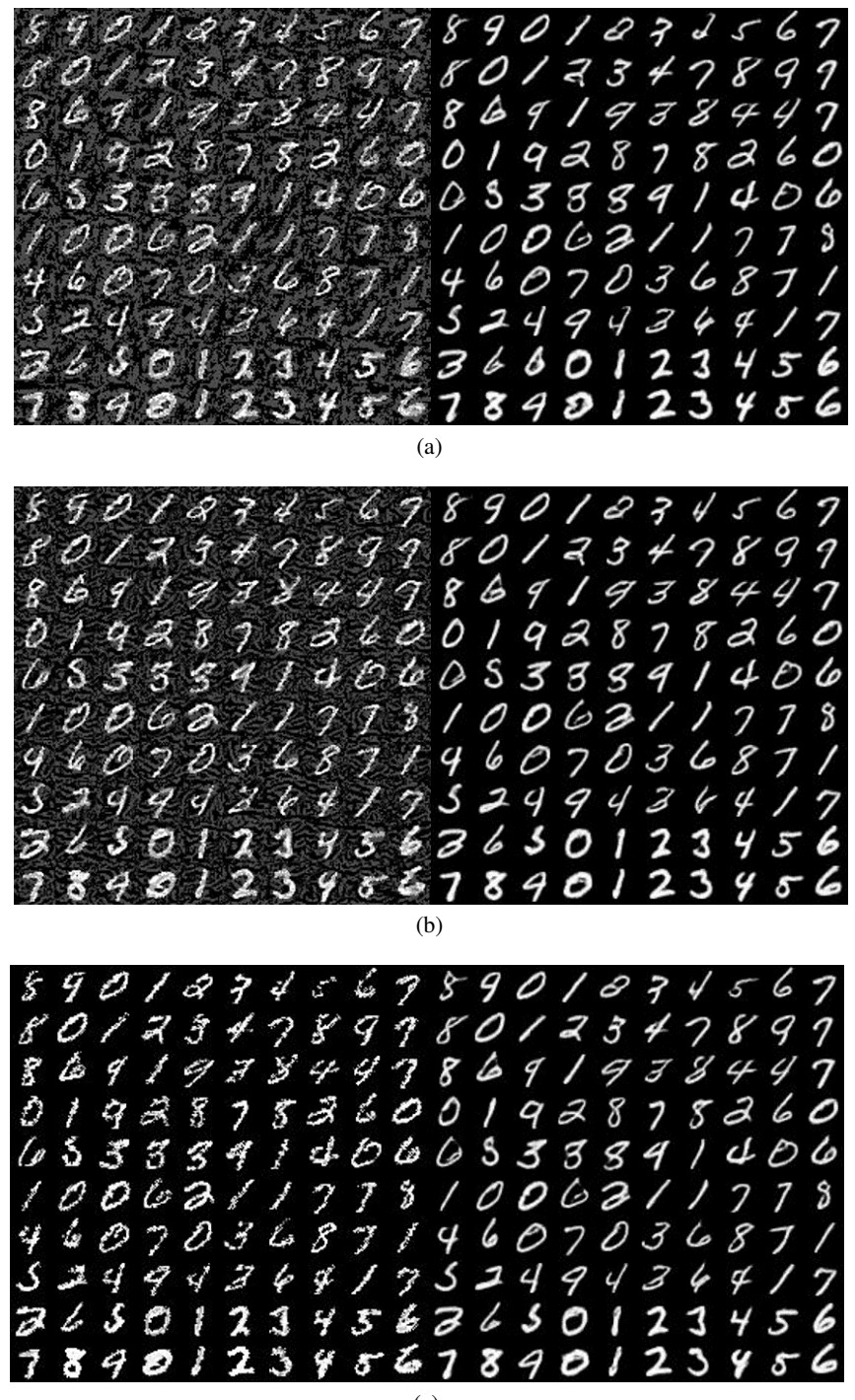

Figure 9: Example results of our method on MNIST. The left part shows adversarial examples generated by different attacks: FGSM attack with $\epsilon = 0.3$ (a), PGD attack with $\epsilon = 3$ (b) and CW with $l_2$ norm (c), while the right part shows purified images by AE-GAN$_{+rs}$. The number of iterations for guided search is set to 15.

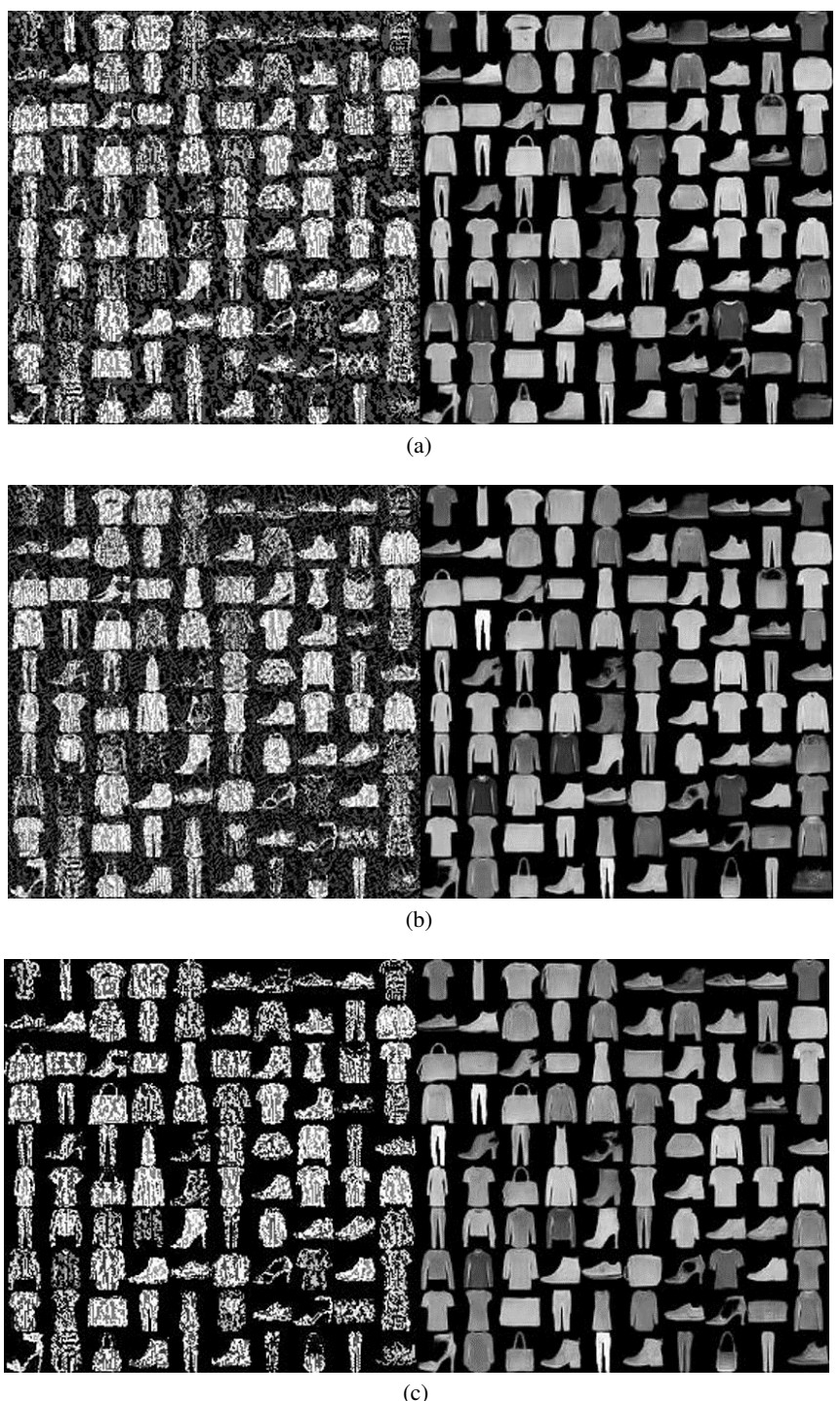

Figure 10: Example results of our method on F-MNIST. The left part shows adversarial examples generated by different attacks: FGSM attack with $\epsilon = 0.3$ (a), PGD attack with $\epsilon = 3$ (b) and CW with $l_2$ norm (c), while the right part shows purified images by AE-GAN$_{+rs}$. The number of iterations for guided search is set to 15.

