# OpenReview forum: "Defense against Adversarial Examples by Encoder-Assisted Search in the Latent Coding Space"
_ICLR.cc/2020/Conference — Reject_

### Official Review · AnonReviewer2 · 2019-10-21
**Official Blind Review #2**

**Rating:** 3

**Review:**

Summary: This paper proposes AE-GAN+sr, an auto-encoder based GAN for equipping neural networks with better defenses against adversarial attacks. The authors evaluate their method on black-box attacks, white-box attacks, and gray-box attacks on MNIST and Fashion-MNIST, and show decent empirical results when compared to baselines.

Decision: Reject. The writing was hard to follow and the experimental evaluations could have been stronger.

Supporting Arguments/Feedback:
- There are a lot of phrases which should be reworded in the text (e.g. “it requires attention to” in paragraph 1, “detect if sample is from a normal distribution” at the end of Section 1, etc.) for better clarity. The writing was pretty hard to follow and the paper would really benefit overall after it has been improved.
- It was hard for me to tell from the text how exactly the models were implemented and trained (see Question #1). Making this point more clear, in either the main text or the Appendix, would be really helpful.
- The performance of AE-GAN+sr as evaluated on MNIST and FashionMNIST did not provide convincing evidence that it outperformed baselines such as Defense-GAN or adversarial training, though I did appreciate the authors’ additional analysis on the tradeoffs between computational cost and performance for the Defense-GAN. The results would also have been more compelling had the authors evaluated their method on more complex datasets such as CIFAR-10.

Questions:
- It wasn’t clear to me how the encoder-assisted search process (Section 3.4) works in practice. When you use gradient descent to find the best encoding for a corrupted input, how many steps do you need to take? Does this step happen while the autoencoder and GAN are being trained, or are those pre-trained and you just take additional gradient steps in the latent space?
- Given that the BiGAN also incorporates an encoder (and they were also benchmarked against in the experiments), where do the advantages of the AE-GAN+sr come from? I think making this point in the text would also be helpful as well.


**Experience Assessment:**

I do not know much about this area.

**Review Assessment: Checking Correctness Of Derivations And Theory:**

I assessed the sensibility of the derivations and theory.

**Review Assessment: Checking Correctness Of Experiments:**

I assessed the sensibility of the experiments.

**Review Assessment: Thoroughness In Paper Reading:**

I read the paper at least twice and used my best judgement in assessing the paper.

---

> ### Author Response · Authors · 2019-11-15
> **Response to Review #2  (part 1/2)**
>
> Thanks for giving us valuable advice. We would like to address your points one by one as follows.
>
> Q1. There are a lot of phrases that should be reworded in the text for better clarity. The writing was pretty hard to follow and the paper would really benefit overall after it has been improved.
> A1. We admit that there are confusing phrases in the original version. We will address your concern by revising the paper thoroughly.
>
> Q2. It was hard for me to tell from the text how exactly the models were implemented and trained. Making this point more clear, in either the main text or the Appendix, would be really helpful.
> A2. Thanks for the suggestion. We would like to provide pseudo-codes for the training and testing stage in Appendix C in our revised version. The testing stage was implemented differently from the training stage. In Sec. 3.3 of the paper, the method for training, denoted as AE-GAN+r, does not include the searching process. When defending against adversarial samples in the inference process, the trained AE-GAN+r becomes AE-GAN+rs (see Sec 3.2 and Sec 3.4) by including the searching process which is denoted as {+s} (see Sec 3.4). More details are given below:
> (1) For the training stage, AE-GAN+r has no searching process and is trained with legitimate samples. As a modified auto-encoder,  AE-GAN+r adds a discriminator to AE and is equipped with an adversarial loss such that the decoder produces realistic images. AE-GAN+r is trained to optimize the min-max loss in Eq.(4) in the manuscript. The training strategy is the same as GANs [1], where the parameters in Encoder-Decoder and the parameters in discriminator are updated alternatively to minimize “Generative Loss + Reconstruction Loss” and “D_Loss” respectively.
> (2) In the inference stage, the AE-GAN+r has been trained and we want to use it to defend against adversarial samples. To get better reconstruction, the searching process is activated, i.e., we use gradient descent to find the best encoding in the latent space for a corrupted input. This defense strategy is denoted as AE-GAN+rs. In practice, we take 15 steps for GD iterations in the searching process, and the learning rate is set to 0.01 (multiplied by 0.5 every 5 steps).
>
> Q3. The performance of AE-GAN+rs as evaluated on MNIST and FashionMNIST did not provide convincing evidence that it outperformed baselines such as Defense-GAN or adversarial training. The results would also have been more compelling had the authors evaluated their method on more complex datasets such as CIFAR-10.
> A3. Thanks for your suggestion. Based on the suggestion, we have added additional results on the Large-scale CelebFaces Attributes (CelebA) dataset in the appendix of the paper. Since it takes more time to prepare the results on CIFAR-10, we will include the results on CIFAR-10 later. The results on CelebA show that AE-GAN+rs is still effective on more complex datasets with large RGB images. For further details, please refer to Appendix G in the revised version. Here we list some results on FGSM ($\epsilon=0.3$) white-box attacks, gray-box attacks as well as black-box attacks.
> ************************************************************************
>                        No Defense        Adv.tr.       Defense-GAN      Our
> White-box       0.0416               0.6119          0.8879               0.9364
> Gary-box         0.0416                ---                 0.8559               0.8897
> Black-box        0.0404               0.6121          0.8517               0.8839
> ************************************************************************
>
> Q4. It wasn’t clear to me how the encoder-assisted search process (Section 3.4) works in practice. When you use gradient descent to find the best encoding for a corrupted input, how many steps do you need to take? Does this step happen while the auto-encoder and GAN are being trained, or are those pre-trained and you just take additional gradient steps in the latent space?
> A4.  Please see the above A2 for the details.
>
> References:
> [1]Goodfellow, I., Pouget-Abadie, J., Mirza, M., Xu, B., Warde-Farley, D., Ozair, S., ... & Bengio, Y. (2014). Generative adversarial nets. In Advances in neural information processing systems (pp. 2672-2680).

---

> ### Author Response · Authors · 2019-11-15
> **Reponse to Review #2 (part 2/2)**
>
> Q5. Given that the BiGAN also incorporates an encoder (and they were also benchmarked against in the experiments), where do the advantages of the AE-GAN+rs come from? I think making this point in the text would also be helpful as well.
> A5. Thanks for the valuable advice. Some related analysis was given in Sec 4.5 (Ablation Study), and here we carry out more experiments to improve the analysis. As shown in Table 3 (page 9) in the original version, although BiGAN also incorporates an encoder, it shows poor reconstruction performance which degrades its defense results.
> *************************************************************
>                                                MNIST                                        FashionMNIST
>                                     BiGAN         AE-GAN+r                 BiGAN           AE-GAN+r
> Clean:                         0.6854          0.9708                        0.5538           0.8267
> FGSM ($\epsilon$=0.3):           0.4593           0.8568                        0.3366           0.6148
> *************************************************************
> The above table is provided with further experimental results in addition to Table 3 of the original version. The accuracies of the target classifier when using BiGAN or AE-GAN+r defense strategy are reported. It could be observed that BiGAN suffers from the problem of poor reconstruction on both clean and perturbed images, and AE-GAN+r (without searching) can increase the accuracy by 30%.
> The discriminator in BiGAN discriminates jointly the data and latent code ( (X, E(x)) versus (G(z), z) ), while the discriminator in AE-GAN is adopted to enhance the generative ability of the decoder such that it can generate realistic images. Thus, the encoder-decoder system is still good at reconstruction in our framework, which is important for the encoder to provide a good initialization for the searching process. BiGAN itself performs badly in purifying a perturbed image, because it may reconstruct an image inconsistent with the input. For a fair comparison, we equip BiGAN with the searching process. As the encoder in BiGAN cannot provide a good start point for the searching process, BiGAN suffers the same problem as Defense-GAN does. In practice, we set the steps=60 for BiGAN and steps=15 for ours, and from Table 1, we can see that AE-GAN+rs still performs much better than BiGAN.

---

### Official Review · AnonReviewer4 · 2019-10-30
**Official Blind Review #4**

**Rating:** 3

**Review:**

The paper combines an auto-encoder (AE) based approach to correct the adversarially perturbed samples with a GAN based approach for the same task. More specifically, the encoder of the AE is used to provide a better initialization for the latent space optimization employed in the Defense-GAN approach.

The authors propose a two-stage inference process. First, the autoencoder is used to detect if an input sample is from the natural image distribution or adversarial distribution. In the latter case, Defense-GAN is employed that uses gradient descent in the latent space, with encoder output as the initialization, to find a natural or non-adversarial counterpart of the input sample. Results are reported for MNIST and F-MNIST that show that the proposed method is computationally cheaper than Defence-GAN.

Questions / Concerns:

 - The methods seem to heavily rely on the autoencoder's ability to detect adversarial samples. The detection performance can be still susceptible to white-box attacks. The methods also relies on the capacity of the auto-encoder to learn good representations/reconstructions. It will be useful to show the performance on more complicated datasets was learning a good AE model is more challenging.
 - The autoencoder is also equipped with an adversarial loss such that the decoder produces realistic images. Is there any assurance that the latent distribution F(x) from the data and the prior p(z) will be similar. It might be useful to explicitly ensure this. The discriminator currently never sees images that re reconstructions of the true data samples. So the decoder behavior could be different for different regions of the input space.
- It is not clear why the proposed method performs better on stringer white-box attacks compared to much weaker black-box attacks.

**Experience Assessment:**

I have read many papers in this area.

**Review Assessment: Checking Correctness Of Derivations And Theory:**

I assessed the sensibility of the derivations and theory.

**Review Assessment: Checking Correctness Of Experiments:**

I assessed the sensibility of the experiments.

**Review Assessment: Thoroughness In Paper Reading:**

I made a quick assessment of this paper.

---

> ### Author Response · Authors · 2019-11-15
> **Reponse to Review #4 (part 1/2)**
>
> We thank you for the comments and suggestions, and answer your questions as follows:
>
> Q1. The methods seem to heavily rely on the auto-encoder's ability to detect adversarial samples. The detection performance can be still susceptible to white-box attacks.
> A1. To defend an adversarial example, our method can always activate the searching process by AE-GAN+rs, i.e., use the code determined by the searching process to reconstruct the example, and then feed the reconstruction into the classifier for the final classification result. To be more efficient, we also design a detection mechanism (in Sec. 4.4) to avoid the unnecessary searching process for the clean examples or the adversarial examples that are affected by small perturbations. If the detection mechanism is used, the adversarial example may cheat the detection step and attack the classifier by the reconstruction from AE-GAN+r. Moreover, as you mentioned in the above comments, the detection mechanism can be susceptible to white-box attacks, which design adversarial examples by attacking the stacked model, i.e., concatenating the classifier with AE-GAN+r as a new classifier. In the following, we conduct experiments to demonstrate that the proposed method is still effective against the above mentioned white-box adversarial attacks.
> *****************************************************************************
> Using MSE as the indicator with threshold=0.01
> *****************************************************************************
>                           AE-GAN+r         AE-GAN+rs     AE-GAN+rs(with detection)     Detection
> FGSM 3.0:          0.691                0.8526                  0.8565                                     100%
> FGSM 1.0:          0.8943              0.9721                  0.9721                                     99.62%
> Cw:                      0.1182             0.7346                  0.7272                                      99.83%
> *****************************************************************************
> Using L1 loss as the indicator with threshold=0.025
> *****************************************************************************
>                          AE-GAN+r         AE-GAN+rs     AE-GAN+rs(with detection)       Detection
> FGSM 3.0:          0.691               0.8526                    0.8565                                    100%
> FGSM 1.0:          0.8943             0.9721                    0.9723                                    100%
> Cw:                      0.1182             0.7346                   0.7275                                     99.82%
> *****************************************************************************
> In the above table, the first column denotes the classification accuracy when using AE-GAN+r to purify the perturbed images under white-box attacks on the stacked model (i.e., concatenating the classifier with AE-GAN+r). That is, no searching process is activated for any input.
> The second column denotes the classification accuracy when using AE-GAN+rs to purify the adversarial examples, i.e., the searching process is activated for every input.
> The third column denotes the classification accuracy when incorporating the detection mechanism to determine whether the searching process is activated for the input.
> Moreover, among the examples that successfully attack the stacked model (i.e., concatenating the classifier with AE-GAN+r), we calculate the percentage of detected ones, as given in the fourth column.
> By comparing the first column and the third column, we can see that the detection mechanism is still effective in detecting the adversarial examples generated by the white-box attacks on the stacked model (i.e., concatenating the classifier with AE-GAN+r). Take the first row for example, when the AE-GAN+r can not diminish the adversarial perturbations for the 1-0.691=30.9% examples, these 30.9% examples can be 100% detected by the detection mechanism (as shown in the last column). Then, they will trigger the searching process and be purified by AE-GAN+rs for higher accuracy. Thus, the final performance of the proposed method (AE-GAN+rs incorporated with detection) on the generated adversarial examples achieves 85.65%, close to AE-GAN+rs (without detection). Therefore, the above white-box attacks didn't affect the performance of the proposed method.
> By comparing the first column and the second column, we can see that the searching process can achieve a higher accuracy on the generated adversarial examples, which demonstrates the effectiveness of the searching process. When the detection mechanism is incorporated, there is no decline in performance. In other words, when the detection helps to save time, it didn’t affect the performance, even under the strongest white-box attacks. We have conducted more experiments to investigate the performance of the detection stage in the appendix in our revised version. For more details, please refer to Appendix F in the revised paper.

---

> ### Author Response · Authors · 2019-11-15
> **Response to Review #4 (part 2/2)**
>
> Q2. The methods also rely on the capacity of the auto-encoder to learn good representations/reconstructions. It will be useful to show the performance on more complicated datasets was learning a good AE model is more challenging.
> A1. Thanks for the suggestions. We agree with the reviewer that our method relies on the capacity of the auto-encoder to learn good representations/reconstructions and it is useful to evaluate the method on more complex datasets. Following the suggestion, we have added results on the Large-scale CelebFaces Attributes (CelebA) dataset in the appendix of the paper. The results show that AE-GAN+rs is still effective on complex datasets with large RGB images. For further details, please refer to Appendix G in the revised version. Here we list some results on FGSM ($\epsilon=0.3$) white-box attacks, gray-box attacks as well as black-box attacks.
> ********************************************************************
>                       No Defense      PGD adv.tr.     DefenseGAN        Our
> White-box       0.0416              0.6119              0.8879                 0.9364
> Gary-box         0.0416              ---                      0.8559                 0.8897
> Black-box        0.0404              0.6121              0.8517                 0.8839
> ********************************************************************
>
> Q3. The autoencoder is also equipped with an adversarial loss such that the decoder produces realistic images. Is there any assurance that the latent distribution F(x) from the data and the prior p(z) will be similar. It might be useful to explicitly ensure this. The discriminator currently never sees images that re reconstructions of the true data samples. So the decoder behavior could be different for different regions of the input space.
> A3.  Thanks for the advice. There is no assurance that the latent distribution F(x) from the data and the prior p(z) will be similar. In AE-GAN, the prior p(z) is a multidimensional normal distribution, while F(x) is derived by the encoder and bounded in $[-1,1]^n$. Based on the advice, we have made the following efforts：(1) Adding a KL-loss between F(x) and p(z) to push F(x) close to p(z). However, the results are not good. One possible reason is that the auto-encoder is trained to self-organize the internal codes of the input patterns, which might not be consistent with the KL loss pushing the codes as a single Gaussian distribution. It may deserve further investigations on the balance between representative power and distribution consistency in the training stage. (2) Construct p(z) to get close to F(x).  We use another GAN (a GAN built in the latent space) to train a generator p(z), which is trained to get close to F(x). However, it is hard to train the two GANs simultaneously and we are not able to make p(z) similar to F(x). Meanwhile, the performance of (2) is not so good as AE-GAN.
> So far, we have not found an effective way to ensure F(x) and p(z) similar. It may be a good topic in future research.
>
>  Q4.  It is not clear why the proposed method performs better on stronger white-box attacks compared to much weaker black-box attacks.
> A4. The main reason for strong defense on white-box attacks lies in the searching process which effectively blocks the gradients. According to the definition of obfuscated gradients in [2], the obfuscated gradients has happened in the proposed AE-GAN+rs, because AE-GAN+rs includes an optimization process when reconstructing the input image, similar to Defense-GAN [1]. Athalye, A., Carlini, N., & Wagner, D. (2018) have proposed a new attack method BPDA [2] to tackle obfuscated gradients, and we also carry experiments on BPDA attacks. We implement the BPDA attack with $\epsilon=0.3$ on the test set of MNIST (same as the setting in Table 1 of the manuscript).  The classification accuracy of the baseline classifier on the MNIST test set is 99% without any attacks. The accuracy of our model against BPDA attacks is 81%, lower than 98.2% by PGD, which shows the BPDA attack is more effective than PGD when attacking our method.  Under BPDA attacks, our method performs worse on white-box than black-box (93.5%).
>
> References:
> [1]Samangouei, P., Kabkab, M., & Chellappa, R. (2018). Defense-gan: Protecting classifiers against adversarial attacks using generative models. arXiv preprint arXiv:1805.06605.
> [2] Athalye, A., Carlini, N., & Wagner, D. (2018). Obfuscated gradients give a false sense of security: Circumventing defenses to adversarial examples. arXiv preprint arXiv:1802.00420.
>
> We hope that this response addresses your concerns, and we would like to answer any other questions you may have.

---

### Official Review · AnonReviewer3 · 2019-11-03
**Official Blind Review #3**

**Rating:** 3

**Review:**

The paper aims to refine DefenseGAN (ICLR 18), where an autoencoder is used to initialize the search for projecting an adversarial examples to the manifold of real examples.

The main contribution is to reduce the computational cost of DefenseGAN, the claim is "by an order of magnitude".

Though the idea is good, I found the contribution to be too incremental for the paper to be accepted:
* The comparison should include the state of the art adversarial training defenses, PGD Adversarial Training (Madry et al; you might want to cite the ICLR 18 paper) and TRADES (Zhang et al., Interpreting adversarially trained convolutional NN, ICML 2019);
* I would consider the Szegedy baseline as obsolete;
* Fig. 4 is unclear: the percentage of adversarial examples detected by the detector, but quid of the false alarms;

There are quite some typos (nosie; iterarion; tabel; unsafety) and missing words. The paper has been hastily written; and it includes one page more than allowed.


**Experience Assessment:**

I have read many papers in this area.

**Review Assessment: Checking Correctness Of Derivations And Theory:**

I assessed the sensibility of the derivations and theory.

**Review Assessment: Checking Correctness Of Experiments:**

I assessed the sensibility of the experiments.

**Review Assessment: Thoroughness In Paper Reading:**

I read the paper at least twice and used my best judgement in assessing the paper.

---

> ### Author Response · Authors · 2019-11-15
> **Response to Review #3**
>
> Thanks for reading our paper thoroughly and the helpful comments. We would like to address your concerns one by one.
>
> Q1. The comparison should include the state of the art adversarial training defenses, PGD Adversarial Training (Madry et al; you might want to cite the ICLR 18 paper) and TRADES (Zhang et al., Interpreting adversarially trained convolutional NN, ICML 2019)
> A1. Thanks for providing the information about the two papers. According to your suggestion, we take PGD adversarial training [1] into comparisons under the same experimental settings. Here we list the comparisons with PGD adversarial training. Due to the time limit, we have not finished the comparisons with TRADES. We will include the comparisons with TRADES later.
> *****************************************************
> On MNIST
> *****************************************************
>                                            None         PGD adv.tr            Our
> White-box      FGSM:       0.144             0.949                  0.984
>                         PGD:          0.007             0.920                  0.982
>                         CW:            0.008             0.773                  0.979
> Black-box:     A/B             0.701             0.971                  0.935
> ******************************************************
> On FashionMNIST
> ******************************************************
>                                           None          PGD adv.tr          Our
> White-box     FGSM:       0.073              0.739               0.804
>                        PGD:          0.028              0.717               0.816
>                        CW:            0.062              0.224               0.767
> Black-box:     A/B            0.227              0.784	          0.603
> ******************************************************
> The results in the above tables indicate that our method can consistently provide effective defense on various attacks, while PGD adversarial training can not generalize to CW attacks. However,  the performances of defenses on the F-MNIST dataset shows noticeably lower than that on MNIST, this is due to the large $\epsilon =0.3$ in the FGSM attack. The qualitative examples in Appendix H can show that $\epsilon=0.3$ represents large perturbations.
>
> Q2. Fig. 4 is unclear: the percentage of adversarial examples detected by the detector, but quid of the false alarms.
> A2.  According to your suggestion, we revised Figure 4 by adding Receiver Operating Characteristic (ROC) curves as well as the Area Under the Curve (AUC) metric. Also, we conduct more experiments on the detection performance and visualize how False Negative Rate (FNR, the percentage of missed diagnosis) and False Positive Rate (FPR, the percentage of false alarms) changes with the threshold increases. The added visualizations on MSNIT are shown in Figure 3 in the revised version, and the results on FashionMNIST are included in Appendix F.  Here we give some examples. For more details, please refer to Figure 3 and Appendix F in our revised version.
> *******************************************************
> Using MSE as the indicator
> *******************************************************
>                                               FPR                                  FNR
>                                                                           $\epsilon=0.1$              $\epsilon=0.3$
> Threshold=0.01          73.37%                        4.34%                 0%
> Threshold=0.015        44.01%                       30.18%                0%
> Threshold=0.02          23.25%                       59.81%                0%
> *******************************************************
> Using L1 distance as the indicator , $\epsilon=0.1$
> *******************************************************
>                                         FPR             FNR
> Threshold=0.02          20.15%          0%
> Threshold=0.025        9.65%            0%
> Threshold=0.03          4.35%            0%
> ********************************************************
> Notation:  $\epsilon$ is the strength of attacks.
>
> Q3. There are quite some typos (nosie; iterarion; tabel; unsafety) and missing words. The paper has been hastily written; and it includes one page more than allowed.
> A3. Thanks for carefully checking our presentation and pointing out our typos. We have carefully revised our paper and improved the quality and readability. To meet the page limit, we tried to revise the paper accordingly but the paper is still a little more than the recommended paper length, but we do not exceed the strict upper limit of the length (10 pages).
>
> References:
> [1] Madry, A., Makelov, A., Schmidt, L., Tsipras, D., & Vladu, A. (2017). Towards deep learning models resistant to adversarial attacks. arXiv preprint arXiv:1706.06083.

---

### Official Review · AnonReviewer1 · 2019-11-04
**Official Blind Review #1**

**Rating:** 6

**Review:**

Authors propose to use a modified autoencoder at the input of a network to ward off adversarial samples by purifying them. The encoder-decoder is trained with a discriminator which can identify whether the input is malicious or normal. A malicious input image is purified with the help of a search procedure in the latent space of the auto-encoder. The gradient based search obtains a latent code that corresponds to the reconstructed image that is closest to the input. The algorithm uses output of the decoder as a starting point in the search. Experiments show a reasonable performance against adversarial attacks on MNIST and F-MNIST images. The proposal seems to be an alternative to the Defence-GAN method of (Samangouei et. al. 2018) where search is computationally expensive - due to multiple starting points and many iterations. The ease of optimization in the proposed method comes at the cost of accuracy (Table 1). However, for a single starting point during search and fewer iterations, it outperforms Defense-GAN by a significant margin (Table 2).

**Experience Assessment:**

I have read many papers in this area.

**Review Assessment: Checking Correctness Of Derivations And Theory:**

I assessed the sensibility of the derivations and theory.

**Review Assessment: Checking Correctness Of Experiments:**

I assessed the sensibility of the experiments.

**Review Assessment: Thoroughness In Paper Reading:**

I read the paper thoroughly.

---

> ### Author Response · Authors · 2019-11-15
> **Response to Review #1**
>
> We thank reviewer #1 very much for their positive and encouraging comments. You are right that the proposed method can be regarded as an alternative to the Defence-GAN method of (Samangouei et. al. 2018) which employed a computational expensive search due to multiple starting points and many iterations. Our method is much faster with comparable performance, and it significantly outperforms Defense-GAN for a single starting point during search and fewer iterations. We have also carefully revised our paper and improved the quality of the paper.

---

### Public Comment · ~Anthony_Wittmer1 · 2019-09-27
**Evaluation questions and obfuscated gradients?**

Hi, I have comments about the experimental results:

1. In table 1, for the proposed model, it seems that the black-box attacks are better than the white-box attacks, which reveals that the defense is obfuscating gradients[1]. The results of Defense-GAN have the same behavior, and Defense-GAN has been broken by [1]. However, for Adv. Tr., the white-box attacks are better than the black-box attacks.

2. In order to check whether obfuscated gradients has happened, BPDA[1] or Nattack[2] is a better choice to evaluate the models.

[1] Obfuscated Gradients Give a False Sense of Security: Circumventing Defenses to Adversarial Examples. ICML 2018
[2] NATTACK: Learning the Distributions of Adversarial Examples for an Improved Black-Box Attack on Deep Neural Networks. ICML 2019

---

> ### Author Response · Authors · 2019-10-10
> **Response to evaluation questions and obfuscated gradients**
>
> Hi Anthony, thanks for your comments.
>
> According to the definition of obfuscated gradients in [2], the obfuscated gradients has happened in the proposed AE-GAN{+sr}, because the proposed method also includes an optimization process when removing the adversarial perturbations in the adversarial examples, similar to Defense-GAN [1]. To investigate the performance of our method against the BPDA attack [2], we use the official implementation of BPDA (https://github.com/anishathalye/obfuscated-gradients) to generate adversarial examples for evaluation. Since our method was implemented in PyTorch, so we rewrite the source code of BPDA in PyTorch to use it for our method. Our implementation is available at https://github.com/Annonymous-repos/attacks-in-pytorch . With all the settings in the source code kept unchanged, we implement the BPDA attack with $\epsilon=0.3$ on the test set of MNIST (same as the setting in Table 1 of the manuscript). The classification accuracy of the baseline classifier on the MNIST test set is 99.6% without any attacks. The accuracy of our model against BPDA attacks is 81%,  lower than 98.2% by PGD, which shows the BPDA attack is more effective than PGD.
>
> On page 12 (Appendix B) of [2], another attack method in “Evaluation A” was presented to attack Defense-GAN [1]. This attack was implemented as “l2_attack” in the source code (https://github.com/anishathalye/obfuscated-gradients). We find that similar to the results reported for Defense-GAN in [2], the adversarial samples generated by l2_attack in our model can mislead the classifier with nearly 100% success. We admit that the adversarial examples exist directly on the projection of the generator/decoder of our model (Generator can directly generate adversarial samples). However, all adversarial examples should be purified by AE-GAN{+sr} to remove the adversarial perturbations before being fed into the classifier. Since the attacking process takes a long time, we compute l2_attack on the first 1000 samples of test set on MNIST, and our method defends the attack with accuracy 92.4%.
>
> Therefore, we admit that there still exist “bad points” in the latent space, which could be used to generate adversarial samples through the Generator/Decoder, but the encoder in our method can encode these samples into “good points” on the manifold of natural images.  As a result, the outputs of encoder are no longer the original bad points. When we start the search process to find the closet reconstruction, the bad points can hardly be found in few steps with the guidance of MSE reconstruction error, where the number of epochs for the search process is set to 15 in our model.
>
> According to the above experiments, our method seems robust against the two attack methods performed for Defense-GAN in [2].
>
> [1]Defense-gan: Protecting classifiers against adversarial attacks using generative models. ICLR2018
> [2]Obfuscated gradients give a false sense of security: Circumventing defenses to adversarial examples. ICML2018

---

> > ### Public Comment · ~Anthony_Wittmer1 · 2019-10-10
> > **Thanks for the reply**
> >
> > Thanks for the reply and additional results. However, I still have some confusion on the results of Adv.Tr., which reveals that the methods based on adversarial training are weaker than the methods based on obfuscated gradients. But actually it is not true[1][2].
> >
> > Why the authors use FGSM adversarial training as the baseline rather than PGD adversarial training (Madry et al. 2018), which is a necessary baseline. I would have expected the authors provide a comparison to Madry et al. (2018) which provides the strongest white-box robustness to date.
> >
> > In addition, have the authors tried to apply the proposed methods on a much larger dataset, such as CIFAR-10?
> > As we know, MNIST is too easy and over used. More importantly, [3] suggests that MNIST is peculiar in that there exists a simple “closed-form” denoising procedure, which leads to similarly robust models without adversarial training. For an average MNIST image, over 80% of the pixels are in {0, 1}, so the adversarial perturbation are easily clipped.
> >
> > [1] Obfuscated Gradients Give a False Sense of Security: Circumventing Defenses to Adversarial Examples. ICML 2018
> > [2] NATTACK: Learning the Distributions of Adversarial Examples for an Improved Black-Box Attack on Deep Neural Networks. ICML 2019
> > [3] Ensemble Adversarial Training: Attacks and Defenses. ICLR 2018

---

> > > ### Author Response · Authors · 2019-11-15
> > > **Response to evaluation questions and obfuscated gradients**
> > >
> > > Thanks for your reply and your suggestions.
> > >
> > > According to your suggestion, we take PGD adversarial training (Madry et al. 2018) into comparisons under the same experimental settings. Here we list the comparisons on MNIST.
> > > *****************************************************
> > >                                            None         PGD adv.tr            Our
> > > White-box      FGSM:       0.144             0.949                  0.984
> > >                         PGD:          0.007             0.920                  0.982
> > >                         CW:            0.008             0.773                  0.979
> > >                        BPDA:         0.000             0.924                  0.810
> > > Black-box:     A/B             0.701             0.971                  0.935
> > > ******************************************************
> > > As shown in the above table,  PGD adv.tr achives better performance than our method under BPDA[1], which is consistent with the conclusions in [1] and [2]. Hope this will help to answer your confusions.
> > >
> > > Meanwhile, we have added additional results on the Large-scale CelebFaces Attributes (CelebA) dataset in the appendix of the paper. Since it takes more time to prepare the results on CIFAR-10, we will include the results on CIFAR-10 later. The results on CelebA show that AE-GAN+rs is still effective on more complex datasets with large RGB images. For further details, please refer to Appendix G in the revised version.
> > >
> > > [1] Obfuscated Gradients Give a False Sense of Security: Circumventing Defenses to Adversarial Examples. ICML 2018
> > > [2] NATTACK: Learning the Distributions of Adversarial Examples for an Improved Black-Box Attack on Deep Neural Networks. ICML 2019

---

### Decision · Program_Chairs · 2019-12-19

**Decision:**

Reject

**Comment:**

The paper proposes a defense for adversarial attacks based on autoencoders that tries to find the closest point to the natural image in the output span of the decoder and "purify" the adversarial example. There were concerns about the work being too incremental over DefenseGAN and about empirical evaluation of the defense. It is crucial to test the defense methods against best available attacks to establish the effectiveness. Authors should also discuss and consider evaluating their method against the attack proposed in https://arxiv.org/pdf/1712.09196.pdf that claims to greatly reduce the defense accuracy of DefenseGAN.